# Mapping global evidence on strategies and interventions in neurotrauma and road traffic collisions prevention: a scoping review protocol

Santhani M Selveindran [1,2] Muhammad Mukhtar Khan,[2,3]
Daniel Martin Simadibrata [4,5] Peter J A Hutchinson,[2,6] Carol Brayne [1,2]
Christine Hill,[1,2] Angelos Kolias,[2,6] Alexis J Joannides,[2,6] Franco Servadei,[7,8]
Andres M Rubiano,[2,9] Hamisi K Shabani[2,10]

For numbered affiliations see end of article.

**Correspondence to**
Dr Santhani M Selveindran;
ss2604@medschl.cam.ac.uk

## ABSTRACT

**Introduction** Neurotrauma is an important global health problem. This 'silent epidemic' is a major cause of death and disability in adolescents and young adults, with significant societal and economic impacts. Globally, the largest cause of neurotrauma is road traffic collisions (RTCs). Neurotrauma and RTCs are largely preventable, and many preventative strategies and interventions have been established and implemented over the last decades, particularly in high-income countries. However, these approaches may not be applicable globally, due to variations in environment, resources, population, culture and infrastructure. This paper outlines the protocol for a scoping review, which seeks to map the evidence on strategies and interventions in neurotrauma and RTCs prevention globally, and to ascertain contextual factors that influence their implementation.

**Methods and analysis** This scoping review will use the established methodology by Arksey and O'Malley. Eligible studies will be identified from five electronic databases (MEDLINE, EMBASE, CINAHL, Global Health/EBSCO and Cochrane Database of Systematic Reviews) and grey literature sources. We will also carry out bibliographical and citation searching of included studies. A two-stage selection process, which involves screening of titles and abstracts, followed by full-text screening, will be used to determine eligible studies which will undergo data abstraction using a customised, piloted data extraction sheet. The extracted data will be presented using evidence mapping and a narrative summary.

**Ethics and dissemination** Ethical approval is not required for this scoping review, which is the first step in a multiphase public health research project on the global prevention of neurotrauma. The final review will be submitted for publication to a scientific journal, and results will be presented at appropriate conferences, workshops and meetings. Protocol registered on 5 April 2019 with Open Science Framework (https://osf.io/s4zk3/).

## INTRODUCTION

Neurotrauma is an important global health problem.[1] [2] Previous studies and systematic reviews, including one from WHO,

### Strengths and limitations of this study

► This will be the first scoping review to identify and quantify the evidence on all types of preventative strategies and interventions for primary, secondary and tertiary prevention of neurotrauma, using all types of literature from both high and low-income and middle-income countries.

► The scoping review uses a well-established, rigorous scoping review methodology with a systematic, iterative search strategy.

► The literature search will be comprehensive, encompassing electronic databases with peer-reviewed literature, as well as a wide range of grey literature sources including government and non-governmental websites.

► A limitation of this review is the language restriction to English, which may result in the potential to miss relevant articles, especially in the grey literature.

estimate that the annual global incidence of neurotrauma is approximately 500–800 per 100 000.[3] [4] This 'silent epidemic' continues to increase and is predicted to surpass many diseases as a major cause of death and disability.[5] Indeed, WHO Global Burden of Disease estimates indicate that neurotrauma accounts for about 11.8% of total global disability-adjusted life years.[3]

Although neurotrauma typically refers to injury to the brain and/or spinal cord, for this review, neurotrauma will focus on injuries to the head alone.

Globally, the largest cause of neurotrauma is road traffic collisions (RTCs).[3] [5] RTCs are the eighth-leading cause of death in the world, and recent WHO analyses estimate that this could become the fifth-leading cause of death by 2030.[6] RTCs are particularly common in low-income and middle-income

BMJ

countries (LMICs) as a result of rapid urbanisation and motorisation with few or no safeguarding measures.[6–8]

Neurotrauma, whether due to RTCs or other causes, poses major medical, public health and societal problems worldwide due to the impact it has on individuals, families, communities, governments and healthcare systems.[9–13]

Despite some country-to-country variability, many studies have shown that neurotrauma is more common in children and young adults.[5 10 14] Neurotrauma is a major cause of death and the leading cause of disability in those under forty years of age, and is predicted to be the third-leading cause of premature death across all ages by 2020.[4 14] Generally, there is a bimodal age-specific neurotrauma incidence with peaks in childhood (0–10 years) and late adolescence/early adulthood (15–36 years).[10 14] In terms of gender, the incidence of neurotrauma is consistently higher in males with ratios ranging from 1.6 to 6.5, particularly in adolescence and early adulthood.[4 5 14]

The demographic impact of neurotrauma in itself has societal and economic implications. As neurotrauma death is highest in males in the age group where most would be the breadwinners within their families, this can result in financial loss or burden.[9] In those who survive, a large proportion would require prolonged hospitalisation, long rehabilitation and suffer long-term disabilities.[15] Deficits associated with neurotrauma are cognitive (impaired attention, poor executive functioning, poor decision-making), physical (impaired motor functioning), behavioural (aggressive behaviour, impulsivity) and emotional (depression, anxiety).[16–18] All this would not only affect their ability to work, but to care for themselves and to participate in society, impacting the lives of relatives and the community around them.[9 19 20] There would be loss of quality of life (intangible costs), for the sufferer and for their caregivers.[12 18]

Economic losses do not only affect families, but also have impact on countries and the world as a whole. The global economic cost of neurotrauma is estimated to be in the region of US$400 billion annually, where most of the aggregate cost of neurotrauma takes the form of indirect costs, resulting from the adverse effect on people's ability to work (loss of productivity).[9 12 21] The direct costs, which refer to all resources consumed within the healthcare sector as a result of neurotrauma, pose a burden on healthcare systems.[9 22]

The management of neurotrauma encompasses prehospital care (lasting for minutes to hours), in-hospital (hours to weeks) and postacute care (weeks to years).[12 14 23 24] Prehospital care involves the correct assessment and efficient treatment at the site of injury or during transfer, and the prompt transport to a formal, appropriately equipped healthcare facility.[12 24] In-hospital care would be the surgical and non-surgical treatment of neurotrauma, including imaging, neuromonitoring and intensive care, whereas postacute care refers to any form of rehabilitative interventions to enable and empower patients to have an increased quality of life, as neurotrauma is a lifelong disorder.[5 23 24]

The lifetime costs of medical treatment for neurotrauma is said to range from US$600 000 to US$1.8 million per case and are almost always borne by the healthcare system.[21 22] In addition to financial constraint, the high burden of neurotrauma may be disproportionate to the resources available for managing the patient load, particularly in LMICs where there are deficiencies in facilities and manpower.[5 14 25]

Therefore, neurotrauma prevention would be more beneficial in that it would not only save lives, but reduce prevalence of disabilities and save costs within and outside the healthcare system.[12] Neurotrauma prevention is limited to measures which target injury occurrence (primary prevention), and involves providing adequate medical response to manage and minimise harm following an injury (secondary prevention), and mitigating the sequelae and reducing consequent disability (tertiary prevention).[12 26 27] These can be applied at societal, community, household and individual levels.[12 13]

Many high-income countries (HICs) have implemented multiple intervention strategies and projects that have contributed significantly to the reduction of neurotrauma.[7] These range from adapting the environment, legislation, safety education and skills training, to strengthening post-trauma response systems and improving access to acute and postacute care.[12 23 28–30] However, these strategies are not often seen in LMICs due to differences in environment, resources, population, culture and infrastructure.[31–33] It is, therefore, necessary that strategies and interventions match contexts in order to effectively reduce the burden of neurotrauma, not only in LMICs but globally.

The objective of this scoping review is to identify and quantify the breadth of evidence on neurotrauma preventative strategies and interventions, provide a descriptive overview of what these are, where they are implemented, and to ascertain contextual factors that influence their implementation. This in turn will be used to inform policy and practice in the area of neurotrauma prevention. As RTCs are a major cause of neurotrauma, this review will delve into RTCs prevention, in addition to neurotrauma prevention as a whole. RTCs will be defined as a collision or incident involving at least one motorised or unmotorised (ie, pedestrian and cyclist) vehicle in motion, on a road to which the public has right of access.[34]

This methodology is appropriate for reviewing a large body of literature to generate an overview of the evidence on this subject, summarise the results and identify research gaps.[35] To date, several reviews have been carried out to identify, examine and study effectiveness of specific preventative strategies and interventions in particular regions or countries.[7 31 36–47] There is also a scoping review on interventions to reduce road traffic injuries, but this was limited to the African continent, and another one specifically on physiotherapy after neurotrauma.[6 48] The value of this scoping review is that it will include all types of evidence on all types of preventative strategies and interventions in both HICs and LMICs.

## METHODS

### Protocol design

This scoping review is informed by the framework proposed by Arksey and O'Malley[35]. This is a six-step framework that includes:

1. Identifying the research question.
2. Identifying relevant studies.
3. Study selection.
4. Charting the data.
5. Collating, summarising and reporting the results.
6. Consulting and translating knowledge.

As this scoping review is part of a wider scoping exercise that includes stakeholder consultation, stage 6 will not be carried out.

### Stage 1: Identifying the research question

The overarching research question, given the purpose of this review is 'What are the global strategies and interventions in neurotrauma and RTCs prevention?' Being part of a wider scoping exercise that seeks to understand contextual issues relating to neurotrauma and RTCs prevention, the following subreview questions were identified:

1. What are the strategies and interventions in neurotrauma and RTCs prevention in LMICs?
2. What are the strategies and interventions in neurotrauma and RTCs prevention in HICs?
3. In what settings are these strategies and interventions carried out (ie, school based/community based)?
4. What are the contextual factors that can affect or influence the implementation of these strategies and interventions?

### Stage 2: Identifying relevant studies

In this stage, we will identify the criteria to be used to select studies that will be included in the review. The Population, Concept and Context mnemonic will be used to determine study eligibility, as follows[49]:

Population: Adults and children, where children will be taken as those below the age of 18 years.

Concept: Any strategy and/or intervention implemented for the prevention of neurotrauma or RTCs. For this review, this would include primary prevention, referring to any measures that eliminate neurotrauma or RTCs; secondary prevention, which will refer to prehospital care systems and any interventions delivered as part of this care; and tertiary prevention, which will be any form of rehabilitative interventions for neurotrauma patients. We will include publications, which describe established preventative strategies and interventions. Studies reporting on intervention outcomes, which relate to neurotrauma and/or RTCs prevention, or on factors influencing implementation will also be included. For RTCs prevention, we will include studies on strategies and interventions that prevent crashes and those that prevent neurotrauma, should a crash occur.

We will exclude the following studies: Papers on strategies and interventions that do not specifically prevent neurotrauma and/or RTCs, studies where strategies and interventions are ill defined, publications describing proposed strategies or interventions, or opinions on an intervention, studies on strategies and interventions without any specific context, papers describing only the prevalence of intervention utilisation without measuring outcomes or exploring factors influencing utilisation, and publications discussing in-hospital management or care of neurotrauma patients.

For publications on rehabilitative strategies and interventions (tertiary prevention), only articles discussing specific approaches that address cognitive, physical, behavioural and emotional rehabilitation will be included. Articles, including pharmacological interventions, will be excluded.

Context: The preventative strategies and interventions would be carried out or delivered in any country (LMIC or HIC) and in any setting (ie, community or healthcare facility).

The following criteria will also be utilised for study eligibility:

Language: Only studies written in English will be included.

Time frame: Searches will begin from 1974 onwards, which was when the World Health Assembly adopted Resolution 27.59, calling for member states to address the major public health problem of RTCs.[29] For databases established after 1974, searches will commence from their inception.

Types of studies: As a scoping review is designed to cover a wide spectrum of literature, there will be no restrictions as to the type of studies that will be included in this review.

The following electronic databases will be searched:

1. MEDLINE (Ovid).
2. EMBASE (Ovid).
3. CINAHL (EBSCO host).
4. Global Health (EBSCO host).
5. Cochrane Database of Systematic Reviews.

An academic librarian was consulted and gave suggestions on key concepts, and text word and Medical Subject Headings searching. The initial search terms were determined by the main author where the concepts 'neurotrauma', 'RTCs' and 'prevention' were each expressed with a list of synonyms. This was supplemented with keywords and phrases from relevant articles retrieved from an exploratory search of MEDLINE and Google Scholar (see online supplementary appendix, which details the search strategy developed for MEDLINE). The search strategy will be used in the other electronic databases, where the search terms will be adapted and modified based on the requirement of the individual database. As scoping review searches are quite iterative, additional search terms may be identified and incorporated into the search strategy.

We will also search grey literature to identify any non-indexed papers. This would include web-based data sources such as Open Grey, Prevention Information and Evidence Library, The Grey Literature Report and The National Institute for Health and Care Excellence.

Where possible, we will search for unpublished studies from websites of governmental, non-governmental or patient-based organisations engaged in neurotrauma or RTCs prevention (ie, Global Alliance of NGOs for Road Safety).

Bibliographies of eligible studies will be examined to identify any original studies not obtained through the searches. Additionally, citation searching of these studies, using Google Scholar will also be carried out.

Search results will be imported to a bibliographical manager which will be used to store references and remove duplicates. Where necessary, duplicates will also be removed manually.

### Stage 3: Study selection

The screening for potentially eligible publications will follow a two-stage process:

Stage 1: The titles and abstract of potentially eligible studies will be screened against selection criteria, and will be categorised as 'include', 'exclude' or 'uncertain'.

Stage 2: Full-text articles will be sourced for those publications under the 'include' and 'uncertain' categories. The full-text articles will be assessed against the selection criteria and the reason for exclusion will be documented.

This process will be carried out by two reviewers independently. Any disagreements regarding publication inclusion will be resolved through discussion between the two reviewers or third party adjudication.

The process of study selection will be reported using the Preferred Reporting Items for Systematic Reviews and Meta-Analyses (PRISMA) flow chart.

### Stage 4: Charting the data

The extraction of data for a scoping review is known as 'charting the data' and should present information about the study that aligns to the objective and questions of the review.

Appropriate data from the eligible studies will be extracted manually using a customised data extraction sheet designed using Microsoft Word. This will be in tabular form and will be piloted on several studies to ensure all relevant results are extracted, pertinent to the research questions. If necessary, the categories will be modified and the extraction sheet revised.

In line with the research questions for this review, the following data will be extracted:

#### Publication detail
► Citation.
► Publication type (published/unpublished).
► Publication source (journal/website).
► Country of origin.
► Funding source.

#### Study characteristics
► Study design/type.
► Aim(s)/objective(s) of study.
► Study location/setting.

#### If systematic review/meta-analyses
► Number of studies included.
► Years of publication of included studies.
► Countries where included studies were conducted.

#### Participant characteristics
► Age/age range.
► Gender.
► Ethnicity.
► Occupation/socioeconomic status.

#### Preventative strategy/intervention
► Description of intervention.
► Delivery of intervention (how and by whom).
► Setting of intervention.
► Length and intensity of intervention.

#### Reported outcome(s)
#### Factors influencing outcome(s)
The charting of data is an iterative process and the extraction fields may change depending on the included studies.

### Stage 5: Collating, summarising and reporting results

The data collected will be presented using two strategies:
1. Evidence mapping to provide a comprehensive and concise descriptive map of the breadth of research on neurotrauma or RTCs prevention. The extracted data will be categorised based on the year of publication, study design, geographical region, type of participants, type of prevention (primary, secondary, tertiary) and type of intervention. Data will be presented in tabular and graphical forms.
2. A narrative summary describing the included studies and how the results relate to the review objective and research questions.

   The exact reporting format cannot be determined until the data is charted, which would depend on the literature found. If possible, a specific framework will be sought to categorise and map the findings. However, the results will be reported using the PRISMA: Extension for Scoping Reviews.[50]

   There will be no assessment of quality as it is beyond the scope of the review.

### Patient and public involvement

Although patients and public are not involved in the design and conduct of this scoping review, it forms part of a larger scoping exercise, which seeks to gather contextual issues and align research priorities. The review will be used to guide questioning during the stakeholder consultation, which is the second component of this exercise. Stakeholders will include policy-makers, law enforcement officials, healthcare staff, and members of patient and advocacy groups, who could have been neurotrauma patients. Comparisons will be made between findings from this review and the stakeholder consultation.

This review will also form part of the report on the scoping exercise that will be sent for publication and

shared with relevant stakeholders and research funders. We will aim to use government and non-governmental websites and facilities to disseminate data to patients and members of the public.

## CONCLUSION

This article presents a protocol for a scoping review, which is a comprehensive, rigorous and transparent methodology. This review, which includes both peer-reviewed and grey literature, will contribute to research in the area of neurotrauma and RTCs prevention by identifying the breadth of preventative strategies and interventions, their contexts and factors influencing implementation. Through this research, policy and practice can be informed, leading to the development and implementation of evidence-based, context-appropriate, feasible strategies and interventions in resource-constrained settings. This review also allows for discerning gaps in knowledge, and to provide recommendations for future research.

This scoping review is the first step in a multiphase research project, which aims to study the epidemiology of neurotrauma caused by RTCs in LMICs and to understand factors amenable to its prevention in these countries.

## ETHICS AND DISSEMINATION

As the scoping review methodology entails collecting, reviewing and synthesising material from publicly available publications, no ethical approval is necessary. The final review will be submitted for publication to a scientific journal. Where possible, results will be presented at appropriate conferences, workshops and meetings, both nationally and internationally.

**Author affiliations**
[1]Institute of Public Health, University of Cambridge, Cambridge, UK
[2]NIHR Global Health Research Group on Neurotrauma, University of Cambridge, Cambridge, UK
[3]Department of Neurosurgery, Northwest School of Medicine and Northwest General Hospital and Research Centre, Peshawar, Pakistan
[4]Faculty of Medicine, University of Indonesia, Depok, Jawa Barat, Indonesia
[5]Faculty of Medical Sciences, Newcastle University, Newcastle upon Tyne, UK
[6]Department of Clinical Neurosciences, Addenbrooke's Hospital, Cambridge, UK
[7]Department of Neurosurgery, Humanitas University and Research Hospital, Milan, Italy
[8]World Federation of Neurosurgical Societies, Nyon, Switzerland
[9]Department of Neurosurgery, Universidad El Bosque, Bogota, Colombia
[10]Neurological Surgery Unit, Muhimbili Orthopaedic Institute and Muhimbili University College of Allied Health Sciences, Dar es Salaam, Tanzania

**Acknowledgements** The authors thank Veronica Phillips for giving advice and feedback on the search strategy. They also thank Sarah Kelly for providing advice on the research questions and overall conduct of the scoping review.

**Contributors** SMS conceived the idea of a scoping review, developed the research questions and methods, and contributed substantially to drafting and editing this protocol. MMK and DMS helped conceptualise the study inclusion and exclusion criteria. PJAH, AK, CB, CH, AJJ, FS, AMR and HKS gave guidance on study design and provided advisory contributions to the development of this protocol. All authors gave approval to the publishing of the protocol manuscript.

**Funding** This research was commissioned by the National Institute of Health Research using Official Development Assistance (ODA) funding (grant number: RG89187).

**Disclaimer** The views expressed in this publication are those of the author(s) and not necessarily those of the NIHR, National Institute for Health Research or the Department of Health.

**Competing interests** None declared.

**Patient consent for publication** Not required.

**Ethics approval** As the scoping review methodology entails collecting, reviewing and synthesising material from publicly available publications, no ethical approval is necessary.

**Provenance and peer review** Not commissioned; externally peer reviewed.

**ORCID iDs**
Santhani M Selveindran http://orcid.org/0000-0002-3734-1189
Daniel Martin Simadibrata http://orcid.org/0000-0002-7512-2112
Carol Brayne http://orcid.org/0000-0001-5307-663X

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
