## [Reviewer comments · BMJ Open]

ARTICLE DETAILS

TITLE (PROVISIONAL)	Mapping global evidence on strategies and interventions in neurotrauma and road traffic collisions prevention: a scoping review protocol
AUTHORS	M Selveindran, Santhani; Khan, Muhammad; Simadibrata, Daniel; Hutchinson, Peter; Brayne, Carol; Hill, Christine; Koliass, Angelos; Joannides, Alexis; Servadei, Franco; Rubiano Escobar, Andres; Shabani, Hamisi

VERSION 1 – REVIEW

REVIEWER	Katherine Susan Brown The George Institute for Global Health, Sydney, Australia
REVIEW RETURNED	29-Jun-2019

GENERAL COMMENTS	Thank you for the opportunity to review this high quality, carefully thought out paper on a very important and relevant issue. The paper is very well written and comprehensive. I have made some minor comments for your consideration. I see one particular strength of this paper as the high level of attention to detail paid to definitions. - The reviewer provided a marked copy with additional comments. Please contact the publisher for full details.
---

REVIEWER	Krzysztof Goniewicz Medical University of Warsaw Polish Air Force Academy
REVIEW RETURNED	24-Jul-2019

GENERAL COMMENTS	It is suggested to the Authors to re-think the paper and resubmit it. More specifically it is recommended to :  i) restructure the manuscript, pros and cons, ii) in-depth describe a methodology to steer feasibility and further implementation (with indicators and scenarios, if possible), iii) highlight whether and how case study would pave the way for further implementation elsewhere, iv) draw conclusions on possible benefits, (optional – not sure if BMJ ask for it for protocols, you can add some baseline instead) v) clearly state research question behind and paper goals. The abstract should be re-written to reflect the actual content of the paper. Keywords: Traffic accidents? – while in title you are using RTC Also please see: https://www.bmj.com/content/322/7298/1320
---

	Some additional copy editing, both to smooth out the areas of prose due to translation(?) and to use translated terms that are consistent with current technical vernacular will also be required. Although this research was submitted as a protocol I think this manuscript should be reviewed and rewrite it in different way to enable the reader to better understand it.
--	--

REVIEWER	Dr. Desmond Kuupiel School of Nursing and Public Health, University of KwaZulu-Natal, South Africa
REVIEW RETURNED	05-Sep-2019

GENERAL COMMENTS	Review report Reviewer: Desmond Kuupiel, MPH, PhD Thanks for giving me the opportunity to review this interesting protocol aiming to map evidence on strategies and interventions in neurotrauma and road traffic collisions prevention. This protocol is generally well written, but I have some comments and suggestions. Title 1. I suggest it should be rephrased as “Mapping global evidence on strategies and interventions in neurotrauma and road traffic collisions prevention: a scoping review protocol” Abstract 1. Mention the electronic databases that will be searched under the methods section. Article summary 1. Under the strengths and limitations of this study, the authors state that “A limitation of this review is the language restriction to the English, which may result in the potential to miss relevant articles from low and middle income countries, especially in the grey literature”. It is not entirely true because there are other studies published in German, Spanish, French, Portugues, etc which are languages spoken in high income countries as well. I suggest you revise it. Stage 1: Identifying the research question 1. What is the main scoping review question? 2. Which mnemonic was used to evaluate the scoping review question? Example, PCC, 3. Please present a table and cite intext clearly defining this study’s population, concept, and context. 4. The terms “Strategies” and “intervention” are not synonyms but it appears they are been used as such in the sub-questions 1,2, and 3. 5. What is this study’s definition of these terms “Strategies” and “intervention”? This can be defined in the table I suggested you provide. Stage 2: Identifying relevant studies 1. The PCC Mnemonic proposed by the Joanna Briggs Institute is been applied here. Although this study tried to define the population, concept, and context under the eligibility criteria, the definition of the concept in its current form lacks clarity. I am still of the opinion that a brief presentation of the PCC in the table as earlier suggested will be helpful. Subsequently, a detailed inclusion and exclusion criteria can be presented. 2. Under the eligibility criteria, your first paragraph should present only the inclusion criteria and the second paragraph, the exclusion criteria. Do not mix the two. 3. Will this study also consider the primordial prevention?
---

	4. A statement is made that “We will exclude the following,... studies on interventions without any specific context...” This study is mapping evidence globally. So how is this possible? 5. Why are keywords such as “Strategies” and “intervention” not included in your keywords? 6. Google is great for searching grey literature. Will you consider this? Stage 3: Study selection 1. “Stage 1: The titles and abstract of potentially eligible studies will be screened against selection criteria, and will be categorized as ‘include’, ‘exclude’ or ‘uncertain’”. Will you handle the studies classified as “uncertain”? 2. “Stage 2: Full text articles will be sourced for those publications under the ‘include’ and ‘uncertain’ categories”. Why “include” and “uncertain” only at this stage” Again, how will you handle the studies classified as “uncertain” at this stage? 3. A statement is made that “Any disagreements regarding publication inclusion will be resolved through discussion between the two reviewers or third party adjudication”. This not explicit enough. Explicitly state how disagreement will be addressed at each stage of the screening process. 4. Will you calculate the inter-rater agreement following full-text screening? Stage 4: Charting the data 1. Will the data extraction be done manually or with the help of electronic software? 2. To avoid bias, it is advisable two independent reviewers undertake the data abstraction and the Kappa static calculated. I suggest you consider this. I could not see your discussion section. Please include a brief discussion section although you do not have results yet. You may use this as a guide https://www.ncbi.nlm.nih.gov/pmc/articles/PMC6617702/ I did not also see your PRISMA-P. Please include this as a supplementary file and cite it intext. Thank you
--	--

VERSION 1 – AUTHOR RESPONSE

RESPONSE TO REVIEWERS COMMENTS

REVIEWER	REVIEWER COMMENT	RESPONSE	REVISION (Page No.)
1	In INTRODUCTION: “Irrespective of cause, neurotrauma poses major medical, public health and societal problems worldwide due to the impact it has on individuals, families, communities, governments and healthcare systems.” -Perhaps add clarification whether the information regarding neurotrauma from this point is	Thank you for your comments. Yes, we may change the structure of this sentence to make that clear.	INTRODUCTION, paragraph 4 (Page 3): Neurotrauma, whether due to RTCs or other causes, poses major medical, public health and

	just RTCs or any mechanism of injury. I imagine it is probably the latter as there would not be as much neurotrauma burden data specific to RTCs, but it would be good to confirm this.		societal problems worldwide due to the impact it has on individuals, families, communities, governments and healthcare systems.
	In METHODS: Study characteristics:  [ ] Study design/type [ ] Aim(s)/objective(s) of study [ ] Study location/setting Would this include the study size if applicable?	As our scoping review is broad, with the purpose of identifying and quantifying evidence, we would not need to include the size of the study.	-
	In METHODS: 1. Evidence mapping to provide a comprehensive and concise descriptive map of the breadth of research on neurotrauma or RTC prevention. The extracted data will be categorized based on the year of publication, study design, geographic region, participants, type of prevention (primary, secondary, tertiary) and type of intervention. Data will be presented in tabular and graphical forms. Does this part include number of participants?	This would just be type of participants-we will revise this in the text. Again, given the aim of this scoping review, we will not include number of participants.	METHODS, Stage 5 (Page 8): The extracted data will be categorised based on the year of publication, study design, geographic region, type of participants, type of prevention (primary, secondary, tertiary) and type of intervention.
2	It is suggested to the Authors to re-think the paper and resubmit it. More specifically it is recommended to : Restructure the manuscript, pros and cons	Thank you for your comments. The structure of the manuscript is in accordance with that established by BMJ Open. We are uncertain what you mean by pros and cons.	-
	In-depth describe a methodology to steer feasibility and further implementation (with indicators and scenarios, if possible)	Our paper is a scoping review protocol where the scoping methodology is informed by a pre-existing framework proposed by	-

		Arksey and O'Malley (2005). The reporting will be in accordance with the the PRISMA Extension for Scoping Reviews (PRISMA-ScR).The overarching purpose of this review is to identify and quantify the breadth of evidence on preventative strategies and interventions for neurotrauma and road traffic collisions, with the aim of identifying gaps and ultimately influencing policy and practice. This has been highlighted in both the INTRODUCTION and CONCLUSIONS.	
	Highlight whether and how case study would pave the way for further implementation elsewhere	Again, this is a scoping review protocol using a scoping methodology. The methodology does not involve reporting a case study. In relation to further implementation elsewhere, the completed scoping review will report on implementation based on the findings.	-
	Draw conclusions on possible benefits, (optional – not sure if BMJ ask for it for protocols, you can add some baseline instead)	The benefit of this review has been indicated in the INTRODUCTION as below: “The value of this scoping review is that it will include all types of evidence on all types of preventative strategies and interventions in both HICs and LMICs.” It is also highlighted in the CONCLUSION section of the protocol.	-
	Clearly state research question behind and paper goals.	The paper goals are clearly highlighted in the CONCLUSION section as below:	-

		“This review, which includes both peer reviewed and grey literature, will contribute to research in the area of neurotrauma and RTCs prevention by identifying the breadth of preventative strategies and interventions, their contexts and factors influencing implementation. Through this research, policy and practice can be informed, leading to the development and implementation of evidence-based, context-appropriate, meaningful and feasible interventions in resource constrained settings. This review also allows for discerning gaps in knowledge, and to provide recommendations for future research.” The research questions are found in the METHODS (Section: STAGE1: IDENTIFYING RESEARCH QUESTION)	
	The abstract should be re-written to reflect the actual content of the paper	The abstract summarizes the content of the paper, highlighting salient points. It is also in the style and format that is recommended by BMJ Open.	-
	Keywords: Traffic accidents? – while in title you are using RTC Also please see: https://www.bmj.com/content/322/7298/1320	Thank you for the article. We will make the necessary change to the keywords.	Keywords: Neurotrauma prevention, Road traffic collisions prevention, Preventative strategies, Preventative interventions, Low and middle income countries,

			High income countries
	Some additional copy editing, both to smooth out the areas of prose due to translation(?) and to use translated terms that are consistent with current technical vernacular will also be required.	We are uncertain what you mean by 'translation' and 'translated terms that are consistent with current technical vernacular'. Whilst we appreciate this scoping review protocol relates to neurotrauma and Road traffic collisions, the mainstay of the article is the scoping review methodology, which follows an established framework. Furthermore, the technical terms pertaining to the subject matter have been clearly defined in the INTRODUCTION. We did not use any foreign language terms in our article.	-
	Although this research was submitted as a protocol I think this manuscript should be reviewed and rewrite it in different way to enable the reader to better understand it.	Please clarify in what way it should be rewritten. This is a scoping review protocol which follows an established methodology and process. While it may stump readers who are unfamiliar with scoping reviews, it is difficult to rewrite 'to enable the reader to better understand it', as the method follows a set procedure for attaining results.	-
3	Title: I suggest it should be rephrased as "Mapping global evidence on strategies and interventions in neurotrauma and road traffic collisions prevention: a scoping review protocol"	Thank you for your comments. We will take this on board.	TITLE (Page 1): Mapping global evidence on strategies and interventions in neurotrauma and road traffic collisions prevention: a scoping review protocol

	Abstract: Mention the electronic databases that will be searched under the methods section.	We will add these in.	ABSTRACT- Methods and Analysis (Page 2): Eligible studies will be identified from five electronic databases (MEDLINE, EMBASE, CINAHL, Global Health/EBSCO and Cochrane Database of Systematic Reviews) and grey literature sources.
	Article summary: Under the strengths and limitations of this study, the authors state that “A limitation of this review is the language restriction to the English, which may result in the potential to miss relevant articles from low and middle income countries, especially in the grey literature”. It is not entirely true because there are other studies published in German, Spanish, French, Portugues, etc which are languages spoken in high income countries as well. I suggest you revise it	This is a fair point, and we will make the necessary revision.	ARTICLE SUMMARY (Page 2): A limitation of this review is the language restriction to English, which may result in the potential to miss relevant articles, especially in the grey literature.
	What is the main scoping review question?	We will include the main question which would be “What are the strategies and interventions in neurotrauma and road traffic collisions prevention globally?”	METHODS, Stage 1 (Page 5): The overarching research question, given the purpose of this review is “What are the global strategies and interventions in neurotrauma”

			and RTCs prevention?"
	Which mnemonic was used to evaluate the scoping review question? Example, PCC,	We will be using the PCC mnemonic, and will include it in our METHODS.	METHODS, Stage 2 (Page 5): The Population, Concept and Context (PCC) mnemonic will be used to determine study eligibility, as follows
	Please present a table and cite in text clearly defining this study's population, concept, and context.	We feel this a matter of personal preference, as not all scoping review protocols present their eligibility criteria this way, nor is it mandatory to do so as part of the methodology. We have discussed these in the text (see STAGE 2 in the METHODS section).	-
	The terms "Strategies" and "intervention" are not synonyms but it appears they have been used as such in the sub-questions 1,2, and 3. What is this study's definition of these terms "Strategies" and "intervention"? This can be defined in the table I suggested you provide.	The definition of 'strategies and interventions' for this study has been explored in the METHODS, under the 'Concept' section.	-
	The PCC Mnemonic proposed by the Joanna Briggs Institute is being applied here. Although this study tried to define the population, concept, and context under the eligibility criteria, the definition of the concept in its current form lacks clarity. I am still of the opinion that a brief presentation of the PCC in the table as earlier suggested will be helpful. Subsequently, a detailed inclusion and exclusion criteria can be presented.	The definition of the Concept has been presented clearly and thoroughly in the text, where these are interventions and strategies for the different levels of prevention for neurotrauma, and road traffic collisions. Please clarify how the concept in its current form lacks clarity.	-
	Under the eligibility criteria, your first paragraph should present only the inclusion criteria and the second paragraph, the exclusion criteria. Do not mix the two	Given the way the eligibility criteria is presented, it would make more sense to discuss it in the way we have done.	-
	Will this study also consider the primordial prevention?	We have used established, universal definitions of neurotrauma prevention, which do not include	-

		primordial prevention. Hence, it is not considered in this study.	
	A statement is made that “We will exclude the following,... studies on interventions without any specific context...” This study is mapping evidence globally. So how is this possible?	If you look at how we have defined ‘Context, you would see that it refers to a country/country-type or specific setting. What we mean by this statement is if a study described an intervention, for example helmets, without discussing it in the context of a particular country or country-type (High-income or low-and middle- income) so it became an article on helmets in general, we would exclude it.	-
	Why are keywords such as “Strategies” and “intervention” not included in your keywords?	The keywords we have used are ‘Preventative interventions’ and ‘Preventative strategies’	-
	Google is great for searching grey literature. Will you consider this?	We have already ample sites/strategies for grey literature searching, but thank you for the suggestion.	-
	Stage 1: The titles and abstract of potentially eligible studies will be screened against selection criteria, and will be categorized as ‘include’, ‘exclude’ or ‘uncertain’”. Will you handle the studies classified as “uncertain”?	We are uncertain what you mean by ‘handle’. The studies considered ‘uncertain’ after title and abstract screening will undergo full text screening and a decision to include or exclude will be made based on that-this has been explained under ‘Stage 2’ of STUDY SELECTION.	-
	“Stage 2: Full text articles will be sourced for those publications under the ‘include’ and ‘uncertain’ categories”. Why “include” and “uncertain” only at this stage” Again, how will you handle the studies classified as “uncertain” at this stage?	We do not see the value in retrieving full texts for papers that have been excluded after title and abstract screening, hence the sourcing of those under ‘include’ and ‘uncertain’ only. Once more, we do not understand what you mean by ‘handle’. Please see	-

		'Stage 2' under STUDY SELECTION.	
	A statement is made that "Any disagreements regarding publication inclusion will be resolved through discussion between the two reviewers or third party adjudication". This not explicit enough. Explicitly state how disagreement will be addressed at each stage of the screening process.	We do not know how explicit you would like us to be. We are merely stating what we intend to do to resolve the issue of reviewer disagreement should it arise. We have looked at many published scoping review protocols, particularly in BMJ Open and found that they have reported this similarly.	-
	Will you calculate the inter-rater agreement following full-text screening?	No. Please see above response.	-
	Will the data extraction be done manually or with the help of electronic software?	We will be doing so manually using a customised data-extraction sheet and we will add this statement to the METHODS.	METHODS, Stage 4 (Page 7): Appropriate data from the eligible studies will be extracted manually using a customised data-extraction sheet designed using Microsoft ® Word.
	To avoid bias, it is advisable two independent reviewers undertake the data abstraction and the Kappa static calculated. I suggest you consider this.	We appreciate your suggestion, but this is not relevant to all scoping reviews.	-
	I could not see your discussion section. Please include a brief discussion section although you do not have results yet. You may use this as a guide https://www.ncbi.nlm.nih.gov/pmc/articles/PMC6617702/	Thank you for sending your paper through. DISCUSSION sections are not required for protocols published by BMJ Open. Please see the response below from the Assistant Editor: Dear Dr M Selveindran, Please accept our apologies for any confusion caused. We would like to clarify that Discussion sections are not required for study protocols, please see here: https://bmjopen.bmj.com/pages/authors/#protocol	-

		If you have any further questions, please do not hesitate to ask, Best wishes, Amy Branch-Hollis Assistant Editor, BMJ Open BMJ, BMA House, Tavistock Square, London WC1H 9JR E: info.bmjopen@bmj.com	
	I did not also see your PRISMA-P. Please include this as a supplementary file and cite it in text.	We have included all the relevant headings from the PRISMA-P to the best we can as this is a scoping review and not a systematic review protocol, and there are many items which are not applicable to a scoping review protocol. We have also received the following response from the Editorial team: Dear Dr. M Selveindran, Thank you for your email, I hope this finds you well. The editorial request for the inclusion of PRISMA-P alongside your manuscript was sent in error. Please disregard it. We hope this helps, Best wishes, Amy Branch-Hollis Assistant Editor, BMJ Open BMJ, BMA House, Tavistock Square, London WC1H 9JR E: info.bmjopen@bmj.com	

VERSION 2 – REVIEW

REVIEWER	Krzysztof Goniewicz Polish Air Force Academy, Poland
REVIEW RETURNED	10-Oct-2019

GENERAL COMMENTS	The authors have creatively utilized secondary sources and made a good effort to revise the second submission taking into account the reviewer comments.
--

REVIEWER	Dr. Desmond Kuupiel University of KwaZulu-Natal, South Africa
REVIEW RETURNED	12-Oct-2019

GENERAL COMMENTS	Thank you for giving me the opportunity to review this interesting study protocol. I have carefully reviewed the responses to my comments and suggestions and I am satisfied. I do not have any further comment. I wish you well at the next stage.
---